
# The impact of human activity on anthropogenic dust emission over global semi-arid regions

X. Guan[1], J. Huang[1], Y. Zhang[1], Y. Xie[1], and J. Liu[2]

[1] Key Laboratory for Semi-Arid Climate Change of the Ministry of Education, College of Atmospheric Sciences, Lanzhou University, Lanzhou, 730000, China
[2] School of Mechanical and Instrument Engineering, Xi' an University of Technology, Xi'an 710048, China

*Correspondence to*: J. Huang (hjp@lzu.edu.cn)

**Abstract.** Anthropogenic dust is acknowledged as a product of human activities on disturbed soil, and is generated mainly from sensitive and fragile regions including croplands, pastures, and urbanized regions. In this study, we analyzed the behaviour of anthropogenic dust in semi-arid region of globe, and its relationship to human activities. An obvious peak in the total anthropogenic dust column, much higher magnitude than those of wet regions, was observed in semi-arid regions with population growth rates of more than 11.46%. Four typical semi-arid regions, East China, India, North America and North Africa were selected to explore the local difference in anthropogenic dust production. The population growth rates in these areas were approximately 6.16%, 17.71%, 11.21%, and 29.26%, and the anthropogenic dust levels were 0.17 g m$^{-2}$, 0.38 g m$^{-2}$, 0.10 g m$^{-2}$ and 0.21 g m$^{-2}$, which are higher than the natural dust column burden. The anthropogenic dust column burden is positively correlated with the population and population change, indicating a contribution from human activities to the anthropogenic dust production. Based on the fact that anthropogenic dust can act as warming aerosol, the radiative effect of anthropogenic dust in semi-arid regions can not be ignored and requires further investigation.

## 1 Introduction

It is well acknowledged that anthropogenic activities play an important role in drylands climate change. Salinization, desertification, loss of vegetative cover, loss of biodiversity and other forms of environmental deterioration are partly a result of anthropogenic activities. With rapid economic development, the modern lifestyle has become increasingly rely on the consumption of fossil fuels (Barnett and O'Neill, 2010). The spatial patterns of energy consumption have strongly influenced temperature trends in recent years (Li and Zhao, 2012), especially in the developing countries. The economic policy of most developing countries is an extensive economic model. This type of economic policy always results in a lower efficiency of resource use.

Different scenarios suggest markedly different processes and final outcomes of climate change. Jiang and Hardee (2011) found that main factors influencing anthropogenic effect on emission are economic growth, technological changes and population growth, which cannot be easily simulated using numerical models (Zhou et al., 2010). More recently, there has been some progress towards understanding the implications of population growth, workforce and economic intensity for



various scenarios of environmental change (Huang et al., 2015b). It appears that higher densities of younger workers are strongly correlated with increased energy use (Liddle, 2004), carbon dioxide emissions (Liddle and Lung, 2010; Huang et al., 2014) and energy consumption that is accompanied by the release of heat into the atmosphere in addition to greenhouse gases. Therefore, the anthropogenic heating resulting from energy consumption may have a significant continental-scale

warming effect at mid-high latitudes during the winter, according to model simulations (Zhang et al., 2013). However, our understanding of the influences of human activities on the environment in drylands is still extremely limited (Jiang, 2010).

Previous studies (Huang et al., 2012) have shown that the warming is particularly enhanced during the boreal cold season over semi-arid regions at mid-high latitudes in Northern Hemisphere, these regions are also significant sources of dust aerosols (Huang et al., 2006a; Chen et al., 2010; Ye et al., 2012; Jin et al., 2015). Mineral dust is widely distributed and tends

to have a relatively large optical depth (Huang et al., 2010; Xu and Wang, 2015; Xu et al., 2015), the existing atmospheric dust load is hard to explain by natural sources alone (Tegen and Fung, 1995). The atmospheric dust load that originates from soils distributed by human activities, such as various land-use practices, can increase the overall dust load and in turn affect radiative forcing. Efforts to quantify the relative importance of different types of dust sources and the factors that affect emissions are critical for understanding the global dust cycle, historical and possible future changes in dust emissions (Okin

et al., 2011; Huang et al., 2015a). Therefore, the proportion of dust types will change the radiative forcing in the related areas (Huang et al., 2006b). Moreover, the long-term radiative forcing change will further change the energy and hydrology circulation over some semi-arid regions (Guan et al., 2009; Wang et al., 2010). Such studies indicate that frequent human activities can make a radiative effect on the climate system from other perspectives besides greenhouse gas.

Generally, anthropogenic dust originates predominantly from agricultural practices (e.g., harvesting, ploughing and

20 overgrazing) and changes in surface water (e.g., shrinking of the Caspian Sea, Aral Sea and Owens Lake), as well as urban (e.g., construction) and industrial practices (e.g., cement production and transport) (Prospero et al., 2002). Over the past few decades, a combination of the higher frequency of warmer and dryer winters - springs in semi-arid and semi-wet regions and changes in vegetated land cover due to human activity have likely increased anthropogenic dust emissions over different regions (Mahowald and Luo, 2003). For example, Mulitza et al. (2010) studied the development of agriculture in the Sahel,

which was associated with a large increase in dust emission and deposition in the region, found that dust deposition is related to precipitation in tropical West Africa on the century scale. Huang et al. (2015a) developed a detection method of anthropogenic dust emission and presented a global distribution of anthropogenic dust aerosol. The current consensus is that up to half of the modern atmospheric dust load originates from anthropogenically disturbed soils (Tegen et al., 2004). Such a great proportion of anthropogenic dust will greatly influence the local radiative forcing. Therefore, a study of human

activities on anthropogenic dust column burden over semi-arid regions is critical and necessary for predicting the influence of dust emission, loading and deposition in the future, and how these changes may subsequently affect regional climate change.

In this study, the anthropogenic dust over semi-arid regions is identified by CALIPSO data, and investigated its relationship with human activities. The method used to distinguish the anthropogenic dust from the total dust aerosols is



based on that of Huang et al. (2015a). This paper is organized as follows: Section 2 introduces the datasets used in this study, Section 3 presents the method used to identify the anthropogenic dust aerosols in the semi-arid regions. Section 4 discusses anthropogenic dust emission over global semi-arid region and its relationship to human activities, including a comparison among four different semi-arid regions. All the major findings, followed by a discussion of the radiative effect of anthropogenic dust on regional climate change in semi-arid regions, are given in Section 5.

## 2 Data

### 2.1 The Aridity Index datasets

In this study, we use the aridity index (AI) to classify the different types of regions. The AI is defined as the ratio of the annual precipitation to annual potential evapotranspiration, representing the degree of climatic dryness. The AI dataset used in this study (Feng and Fu, 2013) is based on Climate Prediction Center (CPC) datasets. Drylands are identified as regions with AI values of less than 0.65 and are further classified into hyper-arid (AI < 0.05), arid (0.05 ≤ AI < 0.2), semi-arid (0.2 ≤ AI < 0.5) and dry sub-humid (0.5 ≤ AI < 0.65) (Middleton and Thomas, 1997). Of the four types of drylands, hyper-arid regions are the driest, followed by arid, semi-arid and dry sub-humid regions. The AI dataset was provided by Feng and Fu (2013) and cover the period from 1948 to 2008, with a spatial resolution of 0.5° by 0.5°.

### 2.2 Population data

The population data were drawn from the Gridded Population of the World dataset, version 3 (GPWv3), which is maintained by the Center for the International Earth Science Information Network (CIESIN) and the Centro Internacional de Agricultura Tropical (CIAT). GPWv3 depicts the global population distribution. It is a gridded, or raster, data product that renders global population data at the scale and extent required to illustrate spatial relationships between human populations and the global environment. It aims to provide a spatially disaggregated population layer compatible with datasets from the social, economic and Earth science disciplines. The spatial grid cell resolution is 0.5°×0.5°. The population data estimates are for the years of 1990, 1995, 2000, 2005 and 2010. The population growth rate from 1990 to 2010 is calculated as:

$$P' = \frac{1}{2} \times (\frac{P_{2010} - P_{2000}}{P_{2000}} + \frac{P_{2000} - P_{1990}}{P_{1990}}) \times 100\% \tag{1}$$

where $P'$ is the population growth rate and $P_{1990}$, $P_{2000}$ and $P_{2010}$ are the population densities in 1990, 2000 and 2010, respectively.

### 2.3 Anthropogenic dust detection data

The instrument used to detect anthropogenic dust is the CALIPSO Cloud-Aerosol Lidar with Orthogonal Polarization (CALIOP). CALIOP acquires vertical profiles of elastic backscatter at two wavelengths (532 and 1064 nm) and linear



depolarization at 532 nm from a near-nadir viewing geometry for both day and night (Hu et al., 2007a, b, 2009). The datasets detail the information of the Level 1 backscatter, depolarization ratio, and color ratio profiles along with the Level 2 Vertical Feature Mask (VFM) products and the 5 km aerosol profile products. The CALIPSO algorithm considers the volume depolarization ratios ($\delta_v$) greater than 0.075 to indicate dust (Omar et al., 2009). In the CALIPSO version 3 VFM data,

which is the cloud aerosol discrimination (CAD) algorithm can separate clouds and aerosols based on multi-dimensional histograms of scattering properties (e.g., intensity and spectral dependence), has been used in the identifying process.

## 2.4 Land cover data

The Collection 5.1 MODIS global land cover type product (MCD12C1) in 2011 is used to identify types of anthropogenic dust sources. It includes 17 different surface vegetation types, and was developed based on the data from the International

Geosphere–Biosphere Programme (IGBP) (Friedl et al., 2010), with a spatial resolution of 0.05º ×0.05º. It provides the dominant land cover type and the sub-grid frequency distribution of land cover classes. In the present analysis, croplands, grasslands, cropland mosaics, and urban are the land cover types that are considered as sources of anthropogenic dust. In addition, urban environments are also identified based on the dataset of Global Rural–Urban Mapping Project (GRUMP) v1 with a spatial resolution of 500 m (Schneider et al., 2010). GRUMP is a valuable resource both for researchers studying

human-environment interactions and for applied users working to address critical environmental and societal issues. GRUMPv1 consists of 8 global datasets, they are population count grids, population density grids, urban settlement points, urban-extents grids, land/geographic unit area grids, national boundaries, national identifier grids, and coastlines. These components make the GRUMP v1 provide a raster representation of urban areas.

## 3 Method of detecting the anthropogenic dust aerosol

Recently, Huang et al. (2015a) developed a new method of separating natural and anthropogenic dust and assessing anthropogenic impacts on dust emissions at the global scale using CALIPSO measurements. These authors (Huang et al., 2015a) defined a schematic framework of dust sources and vertical and horizontal transport processes as the foundation to their approach for discriminating anthropogenic dust from natural dust in CALIPSO data, which proceeds in a sequence of four steps. The first step is to detect the total dust load (both natural and anthropogenic). The second step is to determine the

source region from which the dust originates. The third step is to determine the height of the planetary boundary layer (PBL), and the final step is to determine what proportion of the dust, i.e., that subset of the total dust within PBL.

After the anthropogenic dust has been identified by the detection methods described above, the anthropogenic dust column burden was calculated as follows. First, we determined dust extinction coefficient from the "Atmospheric Volume Description", which is used to discriminate between aerosols and clouds in the CALIPSO Level 2 aerosol extinction profile

products. And then the dust extinction coefficients with the highest confidence levels ($|CAD| \geq 70$) (Liu et al., 2010) and quality control flags of QC=0 or QC=1 were selected. The dust optical depth (DOD, $\tau$) were calculated by integrating the



CAD and QC-filtered extinction coefficient of dust aerosols over the height of the dust layer. After calculating the global total DOD ($\tau_t$) and the anthropogenic DOD ($\tau_a$) from the CALIPSO profile products between January 2007 and December 2010, the dust column burdens ($M$) were converted from DOD ($\tau$), which is performed following Ginoux et al. (2001):

$$M = \frac{4}{3}\frac{\rho r_{eff}}{Q_{ext}}\tau = \frac{1}{\varepsilon}\tau \tag{2}$$

where $r_{eff}$ is the dust effective dust radius, $\rho$ is the dust density of dust, $Q_{ext}$ is the dust extinction efficiency, and $\varepsilon$ is the mass extinction efficiency. The formula is also refereed empirical values from Ginoux et al. (2012) and assume the $r_{eff}$=1.2μm, $\rho$=2600 kg m$^{-3}$, $Q_{ext}$=2.5 and $\varepsilon$=0.6m$^2$ g$^{-1}$. This method does not only modify the maximum standard technique developed by Jordan et al. (2010), its derived dust column burden also has a correlation coefficient of 0.73 with the ground-based lidar observation at the Semi-Arid Climate and Environment Observatory of Lanzhou University (SACOL) (Liu et al., 2014),
which indicates its effectiveness in detection anthropogenic dust.

## 4 Results

### 4.1 Anthropogenic Dust emission over global semi-arid regions

Figure 1 shows the global distribution of semi-arid regions along with the mean anthropogenic dust column burden from 2007 through 2010, demonstrating the wide distribution of anthropogenic dust. Most of the areas with high anthropogenic
dust loading are located in the mid to high latitudes of the Northern Hemisphere, such as North China, Mongolia, northern India, central western North America and Sahel. The highest values are generally distributed throughout Eastern China and India. A comparison for the locations of significant concentrations of anthropogenic dust aerosols reveals that the Northern Hemisphere has much more anthropogenic dust than the Southern Hemisphere. Therefore, we selected four geographical regions that encompass semi-arid regions and are influenced by anthropogenic dust in order to quantify the recent changes.
These regions, marked in Fig. 1 are namely, East China, India, North America and North Africa. From a visual inspection of the overlap between the anthropogenic dust distribution and the semi-arid regions, it can be seen that most semi-arid regions coincide with regions of high anthropogenic dust; however, the anthropogenic dust column burdens are different in different semi-arid regions. The semi-arid regions in East China and India appear to have greater amounts of anthropogenic dust than those in North America and North Africa.
Figure 2 displays the regional anthropogenic dust column burden as a function of climatological annual AI during the period of 1948-2004 of globe. The mean AI varies from 0.0 to a maximum of 2.0. Note that the intervals in this figure are non-uniform because they are from the classification standard for classifying different types of regions based on the AI, as defined in Section 2. Semi-arid regions are the transition zones between arid and semi-wet regions; they are defined as areas where the precipitation are less than the potential evaporation, and is characterized by high temperatures (30-45°C) during





the hottest months. According to Huang et al., (2015a), in semi-arid regions, the annual mean precipitation ranges from 200 to 600 mm yr$^{-1}$ and the AI of semi-arid region is between 0.2-0.5. The global semi-arid regions exhibit relatively high peaks in the anthropogenic dust column burden, with AI values ranging between 0.2-0.5. They have also experienced enhanced warming in recent decades (Huang et al., 2012). Recent findings (Guan et al., 2015a, b) indicate that the enhanced warming

in semi-arid region may be predominantly induced by local factors, especially local aerosol levels. Therefore, the anthropogenic dust aerosol effect on the climate change over semi-arid region cannot be ignored.

As found by Huang et al., (2006a), the dust column burden varies with seasons. We present a comparison of the anthropogenic dust column burdens in the summer, spring, autumn, and winter as functions of the mean climatological AI in Fig. 3. The curves of anthropogenic dust column burden are similar in the four seasons. And in all seasons the anthropogenic

dust column burden exhibits a dominant peak in semi-arid regions, with values much larger than those for other types of regions. In semi-arid regions, the total anthropogenic dust column burden is the greatest in the summer, followed by spring, autumn and winter, that may be a result of the frequency of human activities are different in the four seasons. Also construction activity is likely to be greater, and soil is drier and more friable in the seasons expect winter. The agricultural activities are the most frequent in summer, which may explain why the highest values of the anthropogenic dust column

burden are observed during the summer. The seasonal difference in anthropogenic dust indicates that the frequency of human activities has an impact on anthropogenic dust resuspension and column burden.

The semi-arid region as a whole exhibits a dominant peak, with much higher values of the anthropogenic dust column burden than those in other regions. However, the distribution of the anthropogenic dust column burden throughout all semi-arid regions is not uniform; instead, it shows regional difference. There are two categories for dust grid of CALIPSO dataset,

one is the natural dust, with only natural dust in the grid; the other is combined dust, with both anthropogenic and natural dust in the grid. To distinguish these regional differences in dust aerosol levels, we plot the dust column burdens corresponding to natural and combined dust (natural and anthropogenic dust) in the semi-arid regions of globe, North America, East China, North Africa and India in Fig. 4. It is evident that combined dust aerosol column burden is greater than the pure natural dust of globe. Both combined dust aerosol and pure natural dust column burden are the greatest in India,

followed by North Africa and East China, whereas among these regions, the combined dust is greater than pure natural dust, the difference between combined dust and pure natural dust is the largest in North Africa, followed by India and East China. The North American region combined dust burden is a little less than that of the natural dust.

Previous results tend to focus on the emission effect of natural dust aerosol (Huang et al., 2006a; Yi et al., 2011, 2012; Li et al., 2011), the study on the anthropogenic dust is limited. For the areas where both natural and anthropogenic dust are

30 present, we calculated the dust column burden of natural and anthropogenic dust separately as presented in Fig. 5. It shows that anthropogenic dust column burden is greater than that of natural dust. The anthropogenic dust column burden is the greatest in India, followed by North Africa, East China and North America, where among these regions, the natural dust burden is the highest in North Africa, followed by India, North America and East China. The differing rankings of the anthropogenic and natural dust burdens indicate the underlying sources of anthropogenic and natural dust in these regions



are different. Anthropogenic dust production is more strongly influenced by human activities in semi-arid regions, especially in agricultural areas.

Table 1 reports the values of the mean annual anthropogenic and natural dust column burden from combined dust area over the semi-arid regions of East China, India, North America and North Africa. In semi-arid regions of India, the mean anthropogenic dust column burden is 0.38 g m$^{-2}$ and the natural dust column burden is 0.14 g m$^{-2}$; therefore, the percentage of anthropogenic dust in total dust aerosols achieves 73%. This is a relatively large proportion of anthropogenic contributions compared with those in other semi-arid region, indicating that the impact of aerosols on climate change in India is predominantly attributable to anthropogenic dust. The anthropogenic dusts are followed by the North Africa, East China, and North America, the values are 0.21 g m$^{-2}$, 0.17 g m$^{-2}$ and 0.10 g m$^{-2}$. The natural dust column burdens of North Africa, East China and North America are 0.20, 0.03 and 0.06 g m$^{-2}$, whereas the proportion of anthropogenic dust to total aerosol in these four regions are 51%, 85% and 63%, respectively. Different proportions reveal that the greater predominance of anthropogenic dust aerosol observed over certain semi-arid regions may be related to heavier human activity.

**4.2 Population variance in semi-arid regions**

Semi-arid areas, usually adjacent to arid regions, have fragile ecology system that are difficult to support larger population. The semi-arid areas which are mostly cropland, grassland and cropland mosic, are sensitive to the natural change and human activities. Such environment limits number of people for residence. According to the mean population distribution mapped in Fig. 6, the population density in semi-arid regions exhibits a dramatic regional variability. For the four selected semi-arid regions, both India and East China have higher population density (≥250 persons km$^{-2}$), most semi-arid regions of North Africa has a relative lower population density (~40 persons km$^{-2}$), and the population density in semi-arid region of North America is the lowest (~5 persons km$^{-2}$). The regional difference of population indicates influence of human activities are not uniformly distributed in the semi-arid areas.

Figure 7 illustrates the global distribution of population growth rate. The growth rate is related with various factors such as population policies, economic development status and political divisions. Africa exhibits the highest rate of population growth, followed by South America and central Asia. Most high latitudes of Northern Hemisphere exhibit a population density decrease. The population growth rate in western areas of the North America is faster than in eastern areas; a similar spatial pattern of the population growth has occurred in China. The difference between these respective western and eastern areas may have a close relationship with the economic status. The eastern areas of both North America and China, with high population density and low population growth rates are more industrialized than their western parts. In all the selected semi-arid regions, the highest population growth appeared in the North Africa (30-40%), and in the other three regions are in the range of 0-20%. The inconsistent distribution between population growth rate and population density reveals that the regions with the largest population are not necessary to be the regions with the highest population growth rate.



Figure 8 compares the population densities of the four selected regions, it is apparent that India has the highest population density, which reaches almost 260 persons $km^{-2}$, but none of the other regions is more than 80 person $km^{-2}$. For the other regions, the population densities from high to low are East China, North Africa and North America. However, the population growth in these four semi-arid regions, as is illustrated in Fig. 9, is not proportional to their average population densities. The

population growth rate appears to be the highest in North Africa, followed by India, North America and East China. The highest rate of population growth in North Africa has achieved almost 30%, and the lowest rate of population growth is about 8% in East China.

The detailed population densities and population growth rates based on the population densities in 1990, 2000 and 2010 have been illustrated in Table 2, it shows that India has the highest population density of 251 persons $km^{-2}$ with a growth rate

of 17.71%, its population density is much higher than the second largest in East China, East China has a population of 45 persons $km^{-2}$, with a population growth rate of 6.16%. The population densities of North Africa and North America are 41 and 19 persons $km^{-2}$, respectively. However, the population growth rate in North Africa is 29.26%, which is the highest among the four regions, followed by India (17.71%), North America (11.21%), and East China (6.16%). Therefore, the actual population changes are about 2.8, 44.4, 2.1 and 12.0 persons $km^{-2}$ per decade in semi-arid regions of East China, India,

North America and North Africa.

**4.3 The impact of population on the anthropogenic dust column burden**

According to the previous research (Huang et al., 2015a; Liu et al., 2015), anthropogenic dust is closely related to human activities. Most semi-arid regions locate in the anthropogenic dust areas. Population variance may influence the levels of anthropogenic dust aerosols. In order to avoid the greater difference of grid numbers in different range of population density,

we plotted the total grid number of different population density in semi-arid region with anthropogenic dust of globe (Fig.10). It shows that the grid number of 0-10 persons $km^{-2}$ is much more than the other population densities, almost 7 times larger than that of higher population densities, most of these points locate in the adjacent of semi-arid region. These points have a small proportion of anthropogenic dust area because of rear population. Therefore, those points with the population density between 0-10 persons $km^{-2}$ in semi-arid region are removed in the following analysis.

Figure 11 shows the global mean anthropogenic dust column burden in semi-arid region as a function of population density over four major land types. Statistically, the mean anthropogenic dust burden increases along the increasing population density in cropland (blue line), cropland mosaics (black line) and urban (red line). This positive relationship is most obvious in the croplands where more human activities are involved than the other regions. The anthropogenic dust burden over grassland (green line) is remain unchanged with the population density. Furthermore, an obvious increasing in

anthropogenic dust takes place when the population density is 90-100 persons $km^{-2}$ and reveal that the anthropogenic dust might be more sensitive to the region with larger population density. The difference of anthropogenic dust over different land cover indicates that the land type experiences more human activities, the more anthropogenic dust aerosol will be produced.





Figure 12 is the mean anthropogenic dust column burden as a function of population density. The population varies from 10 to 400 on the X-axis with a non-inform intervals, and the mean anthropogenic dust ranges from 0.15 to 0.35 g m$^{-2}$. The anthropogenic dust starts obvious increase from the population density of greater than 100 persons km$^{-2}$, which is consistent with Fig. 11, and illustrates high population density that greater than 100 persons km$^{-2}$ make significant effect in production

of anthropogenic dust. The standard deviation of anthropogenic dust is the highest in the population of greater than 400 persons km$^{-2}$ and lowest in a population of 25-50 persons km$^{-2}$. Basically, the standard deviation of anthropogenic dust is larger in the high population density. The positive correlation indicates the increasing population density might make a contribution to production of the anthropogenic dust column burden. Figure 13 is the mean anthropogenic dust as a function of population change and aims to explore the sensitive of anthropogenic dust to variance of human activity. The

anthropogenic dust appears obvious increasing from the population change greater than 25 persons km$^{-2}$, and the amplitude of variance becomes larger than population change less than 25 persons km$^{-2}$, with high standard deviation. The positive correlation reveals that the anthropogenic dust increased by population change tend to occur in the case of larger population change and confirm more population will benefit in production of anthropogenic dust in semi-arid region.

## 5 Summary and discussion

In this paper, we focused on building a relationship between variance of anthropogenic dust and the population. It found that the total anthropogenic dust column of globe exhibits an obvious peak in the semi-arid region, which is much higher than those in other regions. Four geographical semi-arid regions, North America, East China, North Africa and India have been chose as our study areas according to their anthropogenic dust levels and the population. The rates of population growth rates were approximately 11.21%, 6.16%, 29.26%, and 17.71% in the semi-arid regions of North America, East China, North

Africa and India, respectively, with anthropogenic dust levels of 0.10 g m$^{-2}$, 0.17 g m$^{-2}$, 0.21 g m$^{-2}$ and 0.38 g m$^{-2}$, respectively, which are higher than the natural dust column. The population density is positively correlated to anthropogenic dust, indicating that the population density contributed to production of anthropogenic dust columns in these semi-arid regions. Meanwhile, the similar correlation between population change and anthropogenic dust occurs in the semi-arid region, also illustrates that both population growth density and population change can be used as dynamic index of human

activities on study the influence of human activities on production of anthropogenic dust column burden. Moreover, the anthropogenic dust accounts for a large proportion of the dust aerosols over semi-arid regions, and act as a warming aerosol, thus may potentially contribute more to the enhanced warming over semi-arid regions than natural dust.

Dust aerosols exert a key impact on the regional radiative forcing over semi-arid regions (Huang et al., 2006b), and are closely related to local climate change (Guan et al., 2015b). Historical statistics reveal that population change occurs in

parallel with economic growth and increases in energy consumption, greenhouse gas emissions and anthropogenic dust. Further studies are needed gain a better understanding of the influence of anthropogenic dust aerosols on climate change in semi-arid regions. Under the current dynamic economic conditions throughout the world, there are still many developing



countries in semi-arid regions that are undergoing extensive economic development or are in the process of transforming from an extensive economic mode to an intensive economic model. Understanding the indirect and semi-direct influences of human activities on climate change is an issue of great importance and requires further study. Developing countries exhibit high rates of population growth, which must be considered when forming economic development strategies. In the developed

5  countries, population growth may also result in increased consumption, higher energy demands and enhanced greenhouse gas production. Therefore, further investigations into the influence of human activities on anthropogenic dust aerosol production, and the consequent impacts on regional climate change in semi-arid regions are needed, with an emphasis on understanding the feedback between regional climate change and societal development with the intent of applying more reasonable policies in the process of economic development.

*Acknowledgements*. This work was jointly supported by the National Basic Research Program of China (2012CB955301), the National Science Foundation of China (41305009, 41575006, 41521004, 41175084), the China 111 project (No. B 13045), and the Fundamental Research Funds for the Central Universities (lzujbky-2015-2, lzujbky-2015-ct03).

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





**Table 1.** Mean dust column burdens in four geographical semi-arid regions (g m$^{-2}$).

| Region | Anthropogenic dust | Natural dust |
|---|---|---|
| East China | 0.17 | 0.03 |
| India | 0.38 | 0.14 |
| North America | 0.10 | 0.06 |
| North Africa | 0.21 | 0.20 |





**Table 2.** Mean population densities/growth rates in four geographical semi-arid regions.

| Region | Mean population density (persons km$^{-2}$) | Growth rate (%) |
|---|---|---|
| East China | 45.28 | 6.16 |
| India | 250.84 | 17.71 |
| North America | 19.18 | 11.21 |
| North Africa | 41.14 | 29.26 |




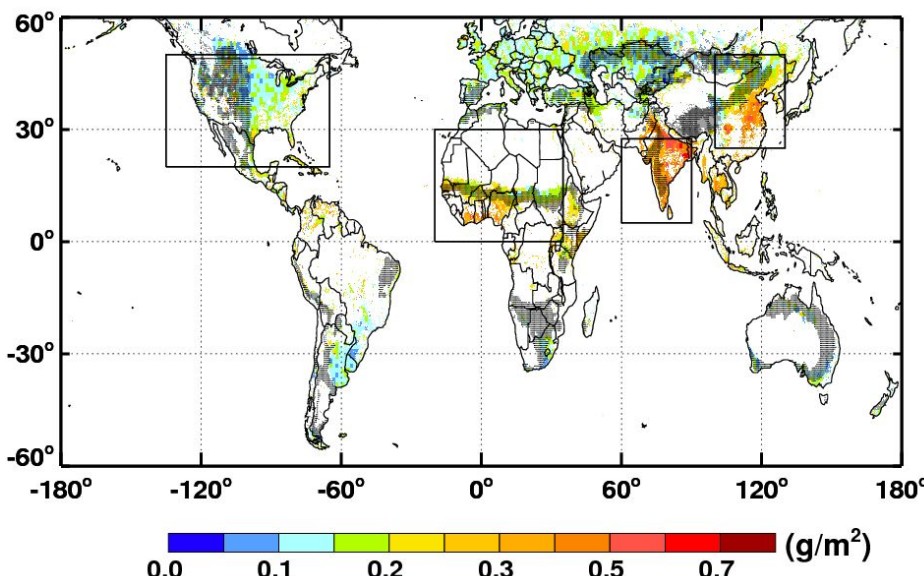

**Figure 1.** Global distribution of mean anthropogenic dust column burden from 2007 to 2010, which is given by the color scale. The gray hatching indicates semi-arid regions.

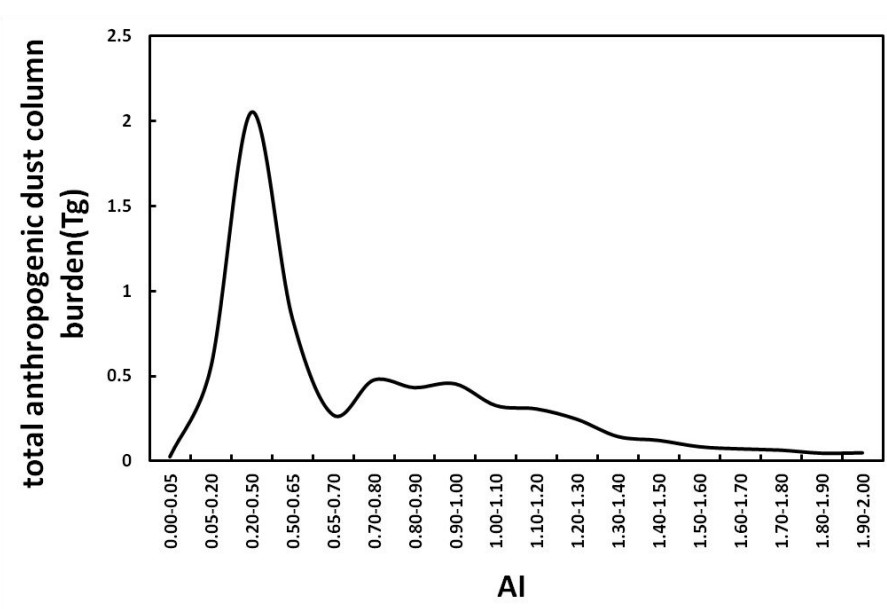

**Figure 2.** Total global anthropogenic dust column burden as a function of the climatological mean AI.





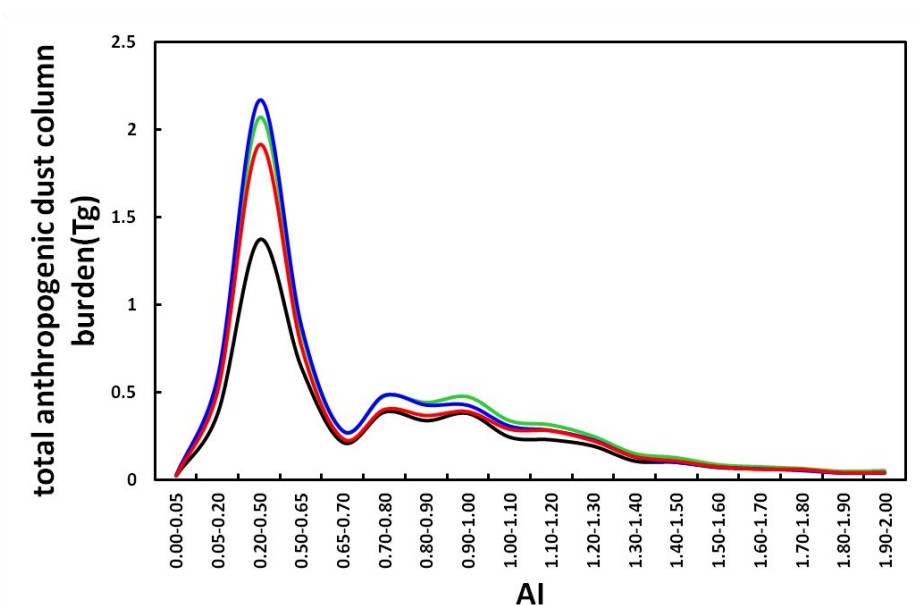

**Figure 3.** Comparison of the global dust column burdens in the spring (green), summer (blue), autumn (red) and winter (black) as functions of the climatological mean AI.





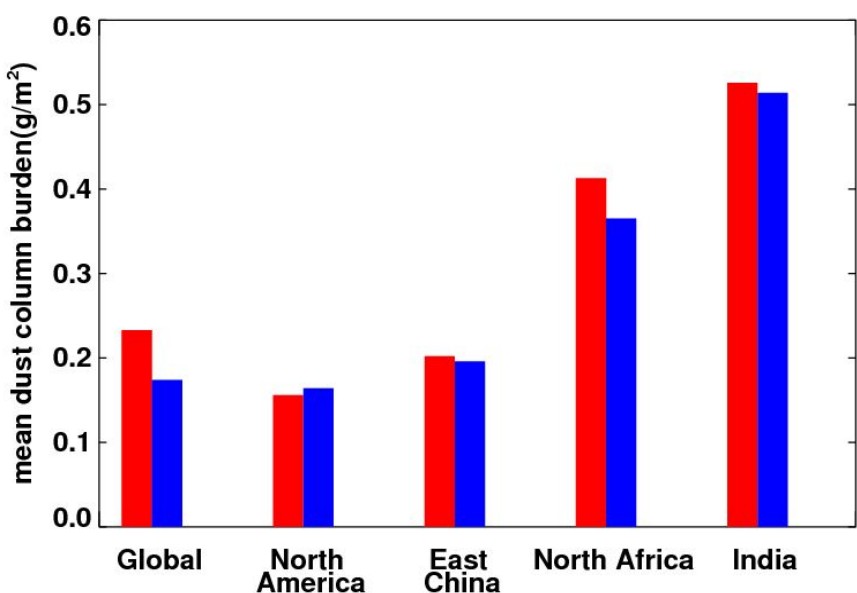

**Figure 4.** Mean dust column burden of anthropogenic dust (red) and combined dust (blue) in the four geographical semi-arid regions.





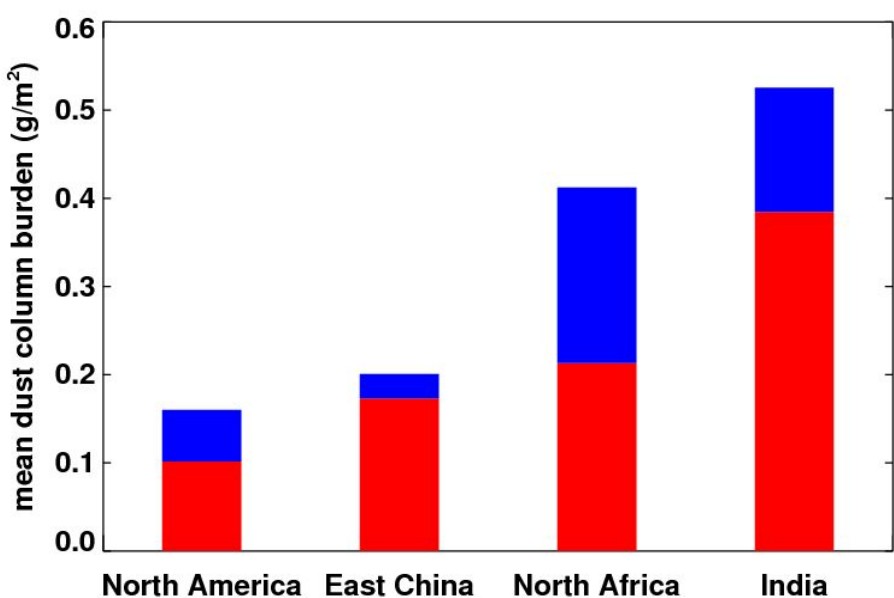

**Figure 5.** Mean anthropogenic (red) and natural (blue) dust column burdens from combined dust regions in the four geographical semi-arid regions.





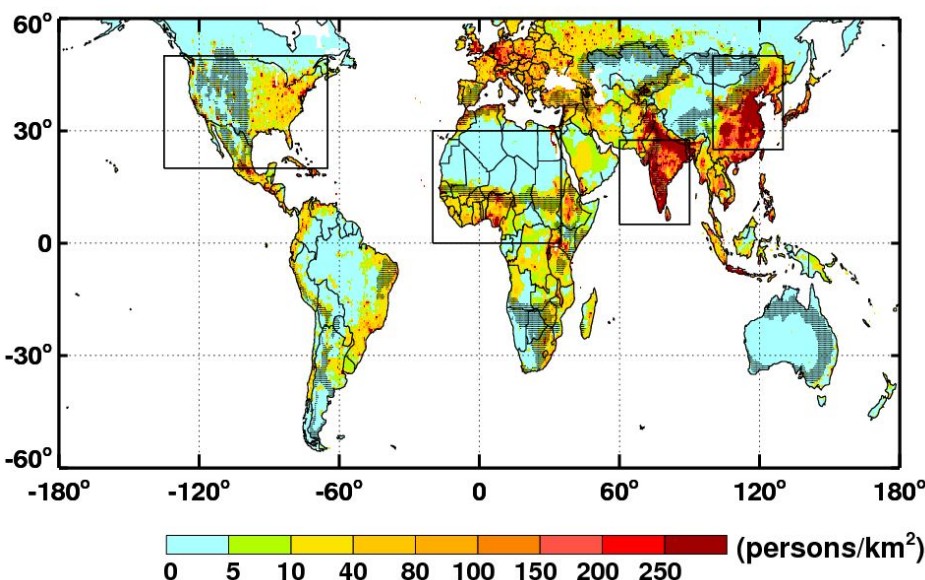

**Figure 6.** Global distribution of mean population density.



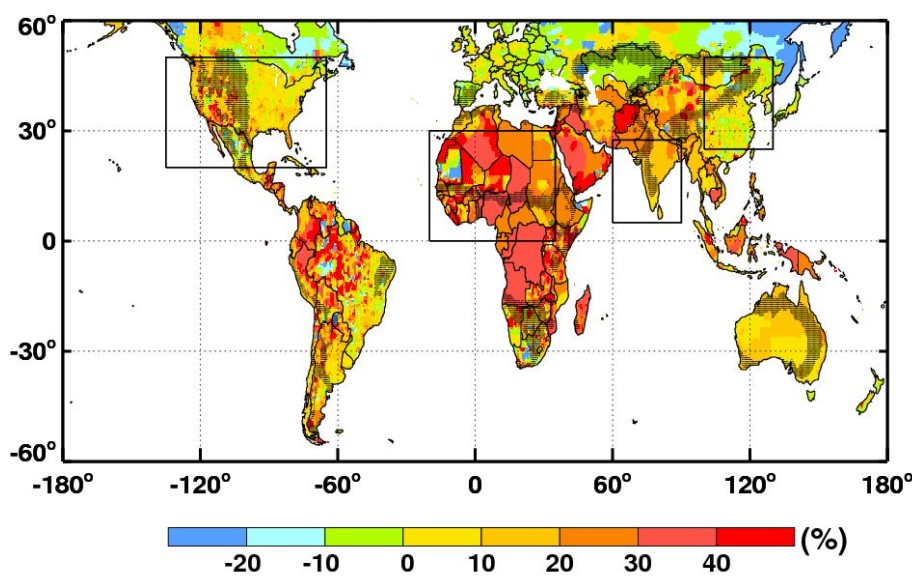

**Figure 7.** Global distribution of mean growth rate in population density.





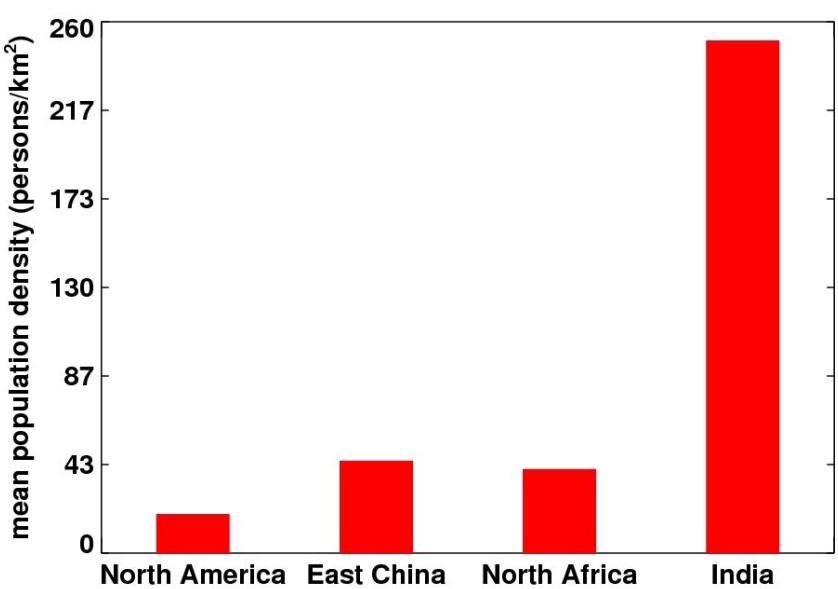

**Figure 8.** Mean population density in the four geographical semi-arid regions.



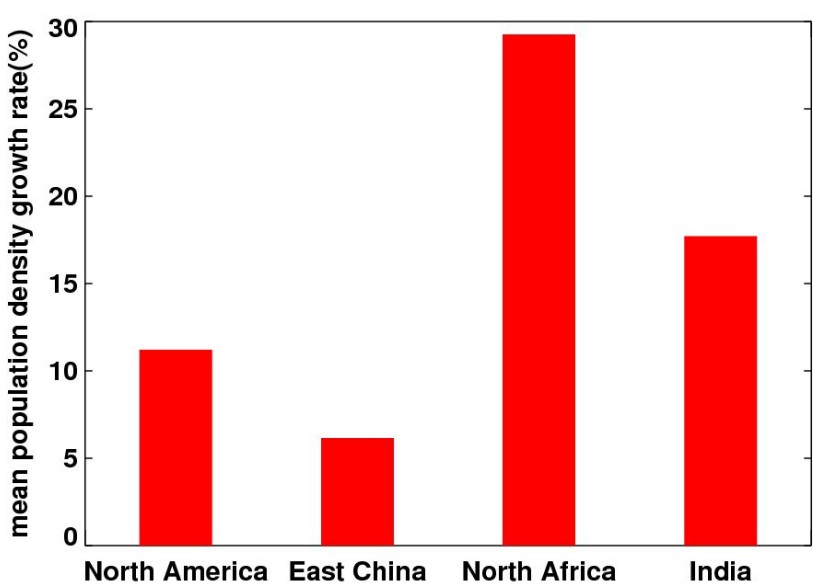

**Figure 9.** Mean population growth rate in the four geographical semi-arid regions.





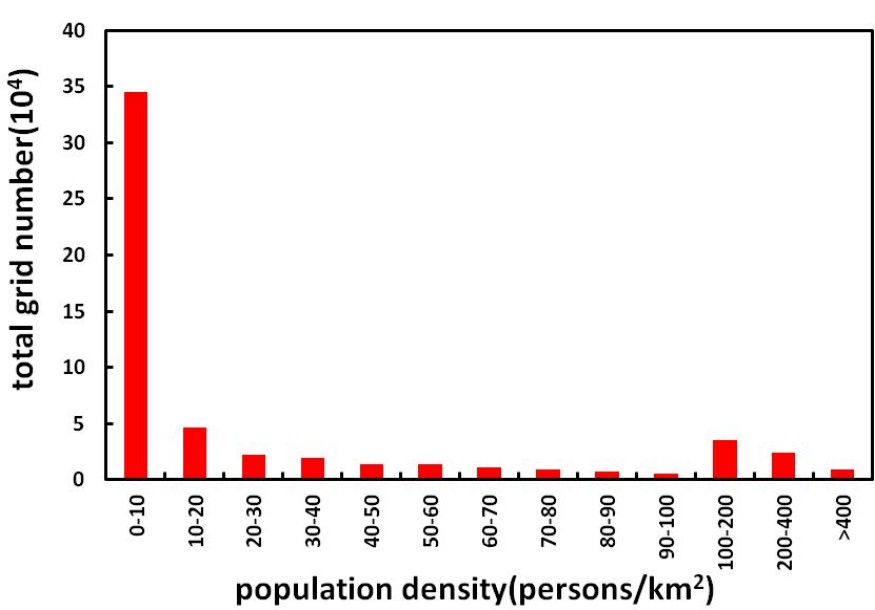

**Figure 10.** Grid number of different population densities in global semi-arid regions with anthropogenic dust.





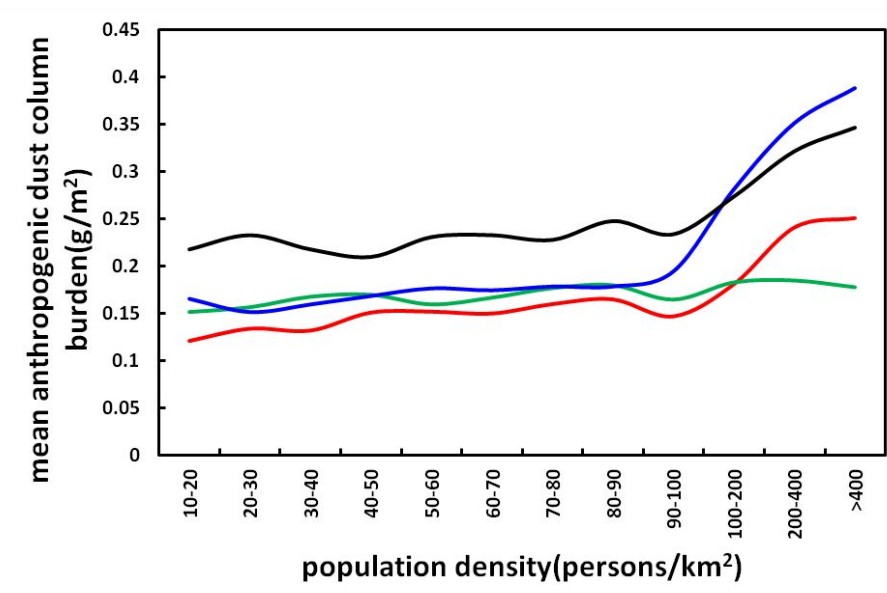

**Figure 11.** Global mean anthropogenic dust column burden as a function of population density in semi-arid regions of Urban (red), Grasslands (green), Croplands (blue) and Croplands Mosaics (black).



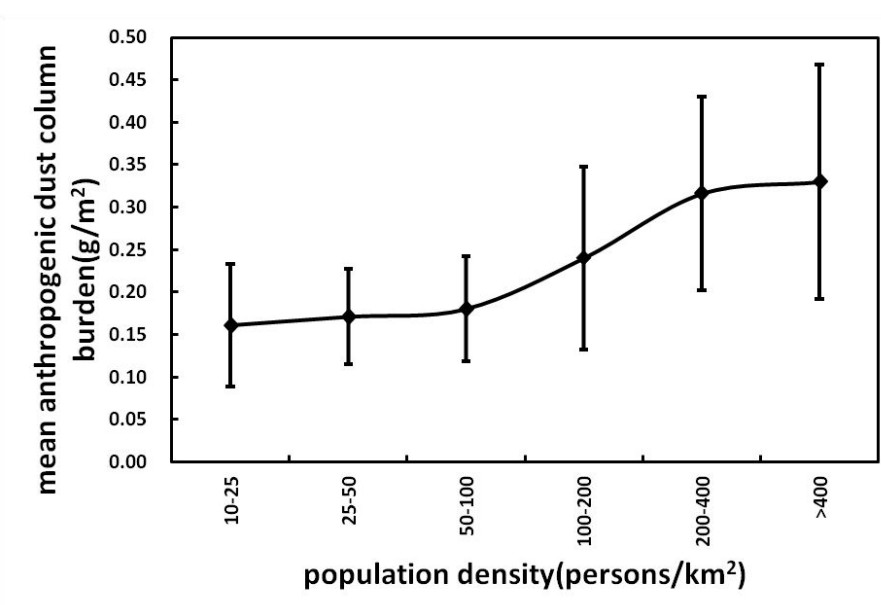

**Figure 12.** Mean anthropogenic dust column burden changes as a function of population density.

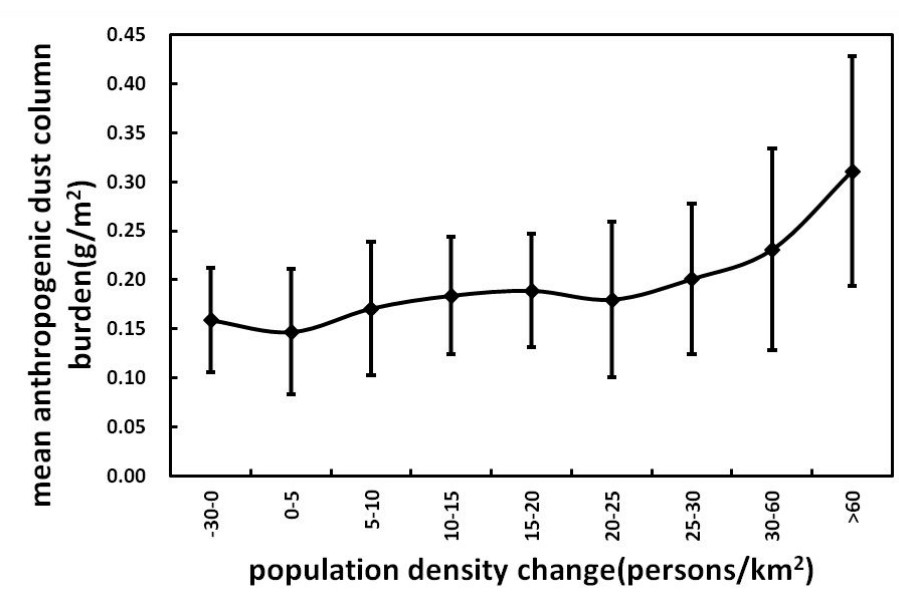

**Figure 13.** Mean anthropogenic dust column burden as a function of population density change.