# Peer review of "The relationship between anthropogenic dust and population over global semi-arid regions"

_Atmospheric Chemistry and Physics, 2015_

## Referee Comment (RC1) · Anonymous Referee #1 · 13 Feb 2016

General comments:

It is known that human activities have impacts on anthropogenic aerosol emissions, but few studies analyze this problem because there are many contributing factors and technical constraints. The authors of this paper employ a state-of-the-art algorithm to distinguish human-induced dust aerosols from CALIPSO satellite observations, and study the relationship between anthropogenic dust burden and population density (growth rate) over various land covers. This paper deserves to be published, but several problems need to be addressed. More discussion, analysis, and references are considered necessary to be added to better support the conclusions.

Major issues:

1. The title is better changed to "The relationship between anthropogenic dust emis-

sion and human activity over global semi-arid regions". The reason for the suggested title change is that the authors barely discussed about the "impacts" but the "relationship". In addition, more in-depth analysis about why such a relationship exists between anthropogenic dust and human activity.

2. The abstract should be revised in order to better reflect the content of this manuscript. The authors should add the temporal ranges of the data used, otherwise the values (i.e. population growth rate, dust burden, etc.) will be meaningless.

3. The introduction part is also deemed insufficient. First, not enough references are provided to support the acclaims. For example, in page 1, line 24-26: "The economic policy of most developing countries is an extensive economic model. This type of economic policy always results in a lower efficiency of resource use." There is no explanation of what is "extensive economic model". And, there is no support (reference or evidence) of why this model "always" results in a lower efficiency of resource use. Similar problems also exist in the manuscript, such as page 1, line 20-22; Also, Page 7, line 15-17: the authors need to add some supporting references to prove that "semi-arid areas have fragile eco-system to support large population" and that "semi-arid area are sensitive to natural change and human activities".

Second, there is not enough discussion of the previous studies about the human impact on anthropogenic dust emission. That is, how do different human activities (i.e. agriculture practice, water use, and industrial practice) practically impact the generation/distribution of anthropogenic dust? This is a critical point in order to understand the variations of human-dust relationship in various regions.

4. Since the four semi-arid regions, namely East China, India, North America, and North Africa, are selected for in-depth study, why the relationships between anthropogenic dust and population index in these regions are not investigated/provided? It is also helpful to show the anthropogenic dust column burden changes as a function of population density in the four regions (Figure 12). These regional evidences are crucial

to support the authors' arguments and thus should be added.

5. A major problem with this manuscript is that quite a few arguments/conclusions derived from the analysis are not considered fully supported by the evidences provided. For instance, in Page 6, line 11-16: the authors argue that the difference in anthropogenic dust in different seasons could be due to the difference in human activities (especially agricultural activities). And, agricultural activities are claimed to be most frequent in summer. Then, why and how do agricultural activities impact the most in summer? Similarly, in Page 7, line 27-29: please explain how the difference in population growth rate closely relates with economic status? In page 8, line 31-33: please explain more about why "the land type experiences more human activities, the more anthropogenic dust aerosol will be produced"? How do you figure out the human activity frequencies?

6. Population density and population change are taken as measurement of human activities. They have a positive relationship with anthropogenic dust in the global semi-arid region. It is better if you can use one figure to show the relationship of anthropogenic dust with population density and population change. In addition, what's the advantage and disadvantage of taking human population density (and variation) as surrogates of human activities? What is the expected impacts on the results?

7. The final problem is with the language. A detail check of the mistakes in grammar and sentence structure is highly recommended.

Minor problems:

* Page 1, line 25: "always" is better changed to "frequently"

* Page 1, line 28: "anthropogenic effect on emission" – emission of what? Aerosols?

* Page 2, line 8: "these regions are . . ." should be "where they are . . ."

* Page 2, line 12: "soils distributed by human activities" should be "soil disturbed by human activities"
* Page 2, line 14: "global dust cycle, historical and possible future changes" should be "global dust cycle, as well as historical and possible future changes"

* Page 2, line 29: "a study of human activity on anthropogenic dust column burden" should be "a study the impact of human activity on anthropogenic dust column burden"

* Page 2, line 33-34: "and investigated its relationship with human activities" should be "and its relationship with human activities is investigated"

* Page 3, line 20: what is "population layer"?

* Page 3, line 24: what is the unit of "population density"?

* Page 3, line26: section "2.3 Anthropogenic dust detection data" is better changed to "2.3 Dust detection data"

* Page 4, line 30: what is "|CAD|>70"? what does it mean?

* Page 5, line 5: "the dust density of dust" should be "the density of dust"

* Page 5, line 7: "This method does not only modify" should be "This method not only modifies"

* Page 5, line 10: "detection" should be "detecting"

* Page 5, line 25: ". . . regional anthropogenic dust . . . of globe" should be ". . . global anthropogenic dust . . ." – I think Figure 2 is the global (not regional) average of anthropogenic dust burden, isn't it?

* Page 6, line 9: it's better to add a legend for Figure 3.

* Page 6, line 12: "that may be a result of " should be "which may be because"

* Page 6, line 28: what do you mean by "emission effect"?

* Page 6, line 34: "differing" should be "different"

* Page 7, line 15: "that are difficult" should be "that is difficult"

* Page 8, line 8-11: please pay attention to the sentence structure. You may consider separate it into several short sentences.

* Page 8, line 23: "rear population" should be "rare population"?

* Page 8, line 27: what is "cropland mosaics"?

* Page 8, line 29: "is remain unchanged" should be "remains unchanged"

* Page 9, line 3: "starts obvious increase" should be "shows obvious increase"

* Page 9, line 4: "make significant effect in production of anthropogenic dust" should be "have significant effect on anthropogenic dust production"

* Page 9, line 9: "the sensitive of" should be "the sensitivity of"

* Page 9, line 10: "appears obvious increasing" should be "shows obvious increase"

* Page 9, line 14: "benefit in production of . . ." should be "contribute to production of . . ."

* Page 9, line 15: "It found that . . ." should be "It is found that . . ."

* Page 9, line 21: "correlated to . . ." should be "correlated with . . ."

* Page 9, line 25: "on study the influence of . . ." should be "to study the influence of . . ."

* Page 17, figure 2 caption: although "AI" is defined in the text, it is still better to give "aridity index (AI)" here for readers who only view the figures.

---

## Referee Comment (RC2) · Anonymous Referee #2 · 24 Feb 2016

The authors aim to explore the relationship between population and anthropogenic dust in semi-arid regions, which has significant implications for local climate change - an important research topic facing the climate change research community. The study clearly reveals that the global semi-arid regions present in average the highest anthropogenic dust burden, and the dust emissions vary substantially across semi-arid regions with different population density and socioeconomic development levels. This paper has great potential for making an important contribution to scholarly discussions on the interactions between human intervention and climate systems at both global and local levels.

Here are some comments and suggestions for the authors to consider in the revision.

My main concern is that the human impacts on dust emissions are not only deter-

mined by the number and growth rate of the population but also affected by the types and intensities of human activities. To choose the four semi-arid regions of different continents and at various socioeconomic development levels for the study of the relations between population density/change and anthropogenic dust burden is a good research design. However, the decision of excluding almost half of the semiarid areas with a population density below 10 persons/m2 from the analysis unfortunately makes the research less robust. The areas excluded are believably dominantly the less populated regions in North America and North Africa, which represents two regimes of human activities and seems to generate very different impacts on anthropogenic dust emissions. While the inclusion of these areas in the analysis of overall interacting patterns may lead to mixed results, one should consider analyzing the relationships in the four regions separately and exploring whether or not there is a common pattern in the relationship between population density and anthropogenic dust burden among all four regions. Even if the resulted relationship varies across regions, it could lead to further analysis of the reasons: why they differ? Is it due to the different levels of aridity, or different types and intensities of human activities? Would the pattern be clearer after controlling AI index, or/and economic level/activity?

Other comments:

In section 4.1, it would be preferable to use "mixed dust" instead of "combined dust" to avoid confusion, particularly when Figure 5 stacks (or combines) anthropogenic and natural dust burden from the "mixed" dust regions.

The sentence of Lines 28-29 on Page 6 can be moved to introduction section, and expressed as a key contribution of this research.

While Figure 4 displays anthropogenic vs. combined (mixed) dust burden, the text on Page 6 talks about the natural vs. mixed dust burden. It should make them consistent.

While Page 7 Line 19 says "both India and East China have higher population density (>= 250 persons km-2) which is also displayed in Figure 6, the other parts of the paper

uses 45 persons/km2 for East China. Is the number in Figure 8 derived from the data of Figure 6? Please explain why.

The last paragraph of Page 7 and Figure 7 is not really relevant and could be removed.

There are some contradictions in texts of the first two paragraphs on Page 8. For instance, it says 8% population increase in East China in the first paragraph but 6.16% in the second; 30% increase in N. Africa in the first paragraph, and 29.26% in the second.

While the paper is generally well written, the second half of the text needs to be improved. In particular, Section 4.2 and 4.3 are not always easy to follow. For instance, what does it mean "Most semiarid regions locate in the anthropogenic dust areas"(Page 8 Line 18)? What is "rear population" (Page 8 Line 23)?

---

## Author Comment (AC1) · 11 Mar 2016

We are grateful for the reviewers' useful advice and comments. They helped us greatly to improve this paper. Our point-by-point responses to the reviewers' comments are listed as follows.

General comments:

It is known that human activities have impacts on anthropogenic aerosol emissions, but few studies analyze this problem because there are many contributing factors and technical constraints. The authors of this paper employ a state-of-the-art algorithm to distinguish human-induced dust aerosols from CALIPSO satellite observations, and study the relationship between anthropogenic dust burden and population density (growth rate) over various land covers. This paper deserves to be published, but several prob-

lems need to be addressed. More discussion, analysis, and references are considered necessary to be added to better support the conclusions.

Part 1:

Major issues:

(1)The title is better changed to "The relationship between anthropogenic dust emission and human activity over global semi-arid regions". The reason for the suggested title change is that the authors barely discussed about the "impacts" but the "relationship". In addition, more in-depth analysis about why such a relationship exists between anthropogenic dust and human activity.

Response: Thank you for your suggestion. The title has been changed to "The relationship between anthropogenic dust emission and human activity over global semi-arid regions." The revised manuscript has also included more description and discussion on the relationship between anthropogenic dust and human activity.

(2)The abstract should be revised in order to better reflect the content of this manuscript. The authors should add the temporal ranges of the data used, otherwise the values (i.e. population growth rate, dust burden, etc.) will be meaningless.

Response: We appreciated this suggestion, the abstract has been revised and the temporal range of data used has also been introduced in the revised manuscript.

(3)The introduction part is also deemed insufficient. First, not enough references are provided to support the acclaims. For example, in page 1, line 24-26: "The economic policy of most developing countries is an extensive economic model. This type of economic policy always results in a lower efficiency of resource use." Three is no explanation of what is "extensive economic model". And there is no support (reference or evidence) of why this model "always" results in a lower efficiency of resource use. Similar problems also exit in the manuscript, such as page 1, line 20-22; Also, Page 7, line 15-17: the authors need to add some supporting references to prove that "semiarid areas have fragile eco-system to support large population" and that "semi-arid area are sensitive to natural change and human activities".

Second, there is not enough discussion of the previous studies about the human impact on anthropogenic dust emission. That is, how do different human activities (i.e.agriculture practice, water use, and industrial practice) practically impact the generation/distribution of anthropogenic dust? This is a critical point in order to understand the variations of human-dust relationship in various regions.

Response: The introduction has been revised, and the questions have been replied separately as below.

(a) in page 1, line 24-26: "The economic policy of most developing countries is an extensive economic model. This type of economic policy always results in a lower efficiency of resource use." Three is no explanation of what is "extensive economic model".

Response: "Extensive economy" is a type of economic growth that depends on high consumption of material resources and energy to a great extent. Its explanation has been added in the revised manuscript.

(b) And there is no support (reference or evidence) of why this model "always" results in a lower efficiency of resource use.

Response: The related references have been added in the revised manuscript. Currently, the high economic growth depends on high consumption of material resources and energy to a great extent, which is a kind of extensive economic growth mode and inevitably encounters the restriction of population, resources, energy, and the pressure of environment, facing a " bottleneck" of the limited resources.

(c) Similar problems also exit in the manuscript, such as page 1, line 20-22;

Response: Similar problems have been fixed in the revised manuscript.

(d) Also, Page 7, line 15-17: the authors need to add some supporting references to prove that "semi-arid areas have fragile eco-system to support large population" and that "semi-arid area are sensitive to natural change and human activities".

Response: More references related to "the semi-arid areas that have fragile ecosystem to support large population" and "the semi-arid area that are sensitive to natural change and human activities" have been added in the revised manuscript.

(e) Second, there is not enough discussion of the previous studies about the human impact on anthropogenic dust emission. That is, how do different human activities (i.e. agriculture practice, water use, and industrial practice) practically impact the generation/distribution of anthropogenic dust? This is a critical point in order to understand the variations of human-dust relationship in various regions.

Response: Thank you for your suggestion. The discussion about human impact on anthropogenic dust emission has been revised. More references about the influence of different human activities on anthropogenic dust have been added in the revised manuscript.

(4) Since the four semi-arid regions, namely East China, India, North America, and North Africa, are selected for in-depth study, why the relationships between anthropogenic dust and population index in these regions are not investigated/provided? It is also helpful to show the anthropogenic dust column burden changes as a function of population density in the four regions (Figure 12). These regional evidences are crucial to support the authors' arguments and thus should be added.

Response: Thanks for your insightful suggestions. The revised manuscript includes the description and discussion over the four typical semi-arid regions, which cover both the relationship between anthropogenic dust and population density, and the relationship between anthropogenic dust and population change. According to your suggestion, we added Figure1 in the revised manuscript to illustrate the relationship between anthropogenic dust aerosol and population density in the four typical semi-arid

regions. Four different semi-arid regions perform different relationships between population density and anthropogenic dust. More description and discussion about the relationship between anthropogenic dust and population density have been stated in the revised manuscript.

(5) A major problem with this manuscript is quite a few arguments/conclusions derived from the analysis are not considered fully supported by the evidences provided. For instance, in Page 6, line 11-16: the authors argue that the difference in anthropogenic dust in different seasons could be due to the difference in human activities (especially agriculture activities). And, agricultural activities are claimed to be most frequent in summer. Then, why and how do agriculture activities impact the most in summer? Similarly, in Page 7, line 27-29: please explain how the difference in population growth rate closely relates with economic status? In page 8, line 31-33: please explain more about why "the land type experiences more human activities, the more anthropogenic dust aerosol will be produced"? How do you figure out the human activity frequencies?

Response: In order to reply the question well, it has been divided into three parts.

(a)For instance, in Page 6, line 11-16: the authors argue that the difference in anthropogenic dust in different seasons could be due to the difference in human activities (especially agriculture activities). And, agricultural activities are claimed to be most frequent in summer. Then, why and how do agriculture activities impact the most in summer?

Response: Spring and summer have the highest anthropogenic dust, which was a conclusion from Huang et al. (2015). They compared the global seasonal distribution of total dust optical depth and found that "the total anthropogenic dust column burden (DCB) is greater in spring and summer than in autumn and winter. This difference is most significant in arid and semi-arid regions. " Summer always has more human activities than the other seasons, both in day and night. It has longer day and indirect induced an increase frequency of human activities.

(b) Similarly, in Page 7, line 27-29: please explain how the difference in population growth rate closely relates with economic status?

Response: Population change reflects the economic status to some extent. For the distribution of economic development in the world, the more developed countries have low population change are, even negative growth; the developing countries usually has positive population growth. It depends on the economic status and style, such as the extensive economic development depends on high consumption of material resources and energy to a great extent; it requires a great number of labor to support development of industries. However, in the developed countries, the high level industrialization needs much less people who has the technology to handle the machines to finish the project that used to need much more people. Therefore, the economic status has the ability to change population growth.

(c) In page 8, line 31-33: please explain more about why "the land type experiences more human activities, the more anthropogenic dust aerosol will be produced"? How do you figure out the human activity frequencies?

Response: Anthropogenic dust aerosol is a type of dust aerosol; it is most originated from exposed land, especially in semi-arid region. Anthropogenic dust aerosol is a result of human activities. According to its sources, anthropogenic dust originates mainly from agricultural practices (harvesting, ploughing, overgrazing), changes in surface water (e.g., shrinking of the Caspian and Aral Sea, Owens Lake), and also from urban practices (e.g., construction), and industrial practices (e.g., cement production, transport) (Prospero et al., 2002). The sentence of "the land type experiences more human activities, the more anthropogenic dust aerosol will be produced" is also been changed to "the land type experiences more human activities, the more anthropogenic dust aerosol may be produced". And Population density and population change have been included in to measure human activities.

(6) Population density and population change are taken as measurement of human

activities. They have a positive relationship with anthropogenic dust in the global semi-arid region. It is better if you use one figure to show the relationship of anthropogenic dust with population density and population change. In addition, what's the advantage and disadvantage of taking human population density (and variation) as surrogates of human activities? What is the expected impact on the results?

Response: Thanks for these insightful suggestions. First, the figures that show the relationship of anthropogenic dust with population density and population change have been combined to one figure. Second, the relationships of anthropogenic dust with population density and population change have been re-organized in the revised manuscript. As we stated in answering the previous question, population and population change have been used as an index of human activities. As an index of human activities, it has both merit and shortcoming. Population-related index has a close relationship with economic development; it is also a result of government policy. However, it has a limitation of scale. Its limitation also can be found in the comparison of four typical semi-arid regions. The traditional agriculture is the most suitable for using the population index, as most people has been limited in the agriculture. The population and its change can greatly impact on anthropogenic dust, which is been greatly reflected in semi-arid region of India. In the semi-arid region of India, traditional agriculture dominated the economic body in selected area, the agriculture anthropogenic dust aerosol exhibited close relations with population density and population change.

Part 2:

final problem is with the language. A detail check of the mistakes in grammar and sentence structure is highly recommended.

Minor problems:

(1) Page 1, line 25:"always" is better changed to "frequently"

Response: Done.

(2) Page 1, line 28:"anthrpogenic effect on emission"-emission of what? Aerosols?

Response: It has been revised as "anthropogenic effect on aerosol emission."

(3)Page 2, line 8: "these regions are ..." should be "where they are ..."

Response: Done

(4) Page 2, line 12:" soils distributed by human activities" should be "soil distributed by human activities"

Response: Done.

(5) Page 2, line 14:"global dust cycle, historical and possible future changes" should be "global dust cycle, as well as historical and possible future changes"

Response: Done.

(6) Page 2, line 29:" a study of human activity on anthropogenic dust column burden" should be "a study the impact of human activity on anthropogenic dust column burden"

Response: Done.

(7) Page 2, line 33-34:"and investigated its relationship with human activities" should be "and its relationship with human activities is investigated"

Response: Done.

(8) Page 3, line 20: what is "population layer"?

Response: It has been changed to "population" to avoid misunderstanding.

(9) Page3, line 24: what is the unit of "population density"?

Response: the unit of population density is persons km-2

(10) Page 3, line26: section "2.3 Anthropogenic dust detection data" is better changed to "2.3 Dust detection data"

[Figure]

Response: Done.

(11) Page 4, line 30: what is "|CAD|>70"? what does it mean?

Response: CAD is cloud aerosol discrimination. |CAD|>70 is a threshold for dust extinction coefficient for the highest confidence level.

(12) Page 5, line 5: "the dust density of dust" should be "the density of dust"

Response: Done.

(13) Page 5, line 7: "This method does not only modify" should be "This method not only modifies"

Response: Done.

(14) Page 5, line 10: "detection" should be "detecting"

Response: Done.

(15) Page 5, line 25: "... regional anthropogenic dust ... of globe" should be "... global anthropogenic dust ..." – I think Figure 2 is the global (not regional) average of anthropogenic dust burden, isn't it?

Response: Done. Figure 2 is the global anthropogenic dust burden.

(16) Page 6, line 9: it's better to add a legend for Figure 3.

Response: The legend for Figure 3 has been added in the revised manuscript.

(17) Page 6, line 12: "that may be a result of " should be "which may be because"

Response: Done.

(18) Page 6, line 28: what do you mean by "emission effect"?

Response: "Emission effect" has been changed to "radiative effect."

(19) Page 6, line 34: "differing" should be "different"

Response: Done.

(20) Page 7, line 15: "that are difficult" should be "that is difficult"

Response: Done.

(21) Page 8, line 8;11: please pay attention to the sentence structure. You may consider separate it into several short sentences.

Response: Thanks. This sentence has been separated into several short ones that are easy to understand.

(22) Page 8, line 23: "rear population" should be "rare population"?

Response: Done.

(23) Page 8, line 27: what is "cropland mosaics"?

Response: Cropland mosaic is a mosaic of less than 60 percentages of cropland in the landscape. Its definition has been added in the revised manuscript.

(24) Page 8, line 29: "is remain unchanged" should be "remains unchanged"

Response: Done.

(25) Page 9, line 3: "starts obvious increase" should be "shows obvious increase"

Response: Done.

(26) Page 9, line 4: "make significant effect in production of anthropogenic dust" should be "have significant effect on anthropogenic dust production"

Response: Done.

(27) Page 9, line 9: "the sensitive of" should be "the sensitivity of"

Response: Done.

(28) Page 9, line 10 "appears obvious increasing" should be "shows obvious increase"

Response: Done.

(29) Page 9, line 14: "benefit in production of ..." should be "contribute to production of ..."

Response: Done.

(30)Page 9, line 15: "It found that ..." should be "It is found that ..."

Response: Done.

(31) Page 9, line 21: "correlated to ..." should be "correlated with ..."

Response: Done.

(32) Page 9, line 25: "on study the influence of ..." should be "to study the influence of ..."

Response: Done.

(33) Page 17, figure 2 caption: although "AI" is defined in the text, it is still better to give "aridity index (AI)" here for readers who only view the figures.

Response: The description of AI has been added in the place for readers to follow the manuscript easily.

Reference:

(1) Huang, J., Liu, J., Chen, B., and Nasiri, S. L.: Detection of anthropogenic dust using CALIPSO lidar measurements, Atmos. Chem. Phys., 15, 11653–11655, doi:10.5194/acp-15-11653-2015, 2015.

(2) Prospero, J. M., Ginoux, P., Torres, O., Nicholson, S. E., and Gill, T. E.: Environmental characterization of global sources of atmospheric soil dust identified with the Nimbus 7 Total Ozone Mapping Spectrometer (TOMS) absorbing aerosol product, Rev. Geophys., 40, 2-1–2-31, doi:10.1029/2000RG000095, 2002.

[Figure]

Fig. 1. Anthropogenic dust probability distributions of semi-arid regions in (a) East China, (b) India, (c) North America, (d) North Africa

---

## Author Comment (AC2) · 11 Mar 2016

We are grateful for the reviewers' useful advice and comments. They helped us greatly to improve this paper. Our point-by-point responses to the reviewers' comments are listed as follows.

**General comments:**

The authors aim to explore the relationship between population and anthropogenic dust in semi-arid regions, which has significant implications for local climate change-an important research topic facing the climate change research community. The study clearly reveals that the global semi-arid regions present in average the highest anthropogenic dust burden, and the dust emissions vary substantially across semi-arid regions with different population density and socioeconomic development levels. His

paper has great potential for making an important contribution to scholarly discussions on the interactions between human intervention and climate systems at both global and local levels.

**Part 1:**

(1) Here are some comments and suggestions for the authors to consider in the revision. My main concern is that the human impacts on dust emission are not only determined by the number and growth rate of the population but also affected by the types in intensities of human activities. To choose the four semi-arid regions of different continents and at various socioeconomic development levels for the study of the relations between population density/change and anthropogenic dust burden is a good research design. However, the decision of excluding almost half of the semiarid areas with a population density below 10 persons km-2 from the analysis unfortunately makes the research less robust.

Response: We appreciated the reviewer's insightful question and agreed that almost half of the semi-arid areas has a population density below 10 persons km-2. The figures 1 and 2 are the revised figures include the population density below 10 persons km-2. They are similar with the primary figures. The new figures in the revised manuscript and related description have been updated in the revised manuscript.

(2) The areas excluded are believably dominantly the less populated regions in North America and North Africa, which represents two regimes of human activities and seems to generate very different impacts on anthropogenic dust emissions. While the inclusion of these areas in the analysis of overall interacting patterns may lead to mixed results, one should consider analyzing the relationships in the four regions separately and exploring whether or not there is a common pattern in the relationship between population density and anthropogenic dust burden among all four regions. Even if the resulted relationship varies across regions, it could lead to further analysis of the reasons: why they differ? Is it due to the different levels of aridity, or different types and
Response: We agree and appreciate the reviewer's suggestion and comment. As the reviewer mentioned, the points with population densities less than 10 persons km-2 are greatly located in North America and North Africa, since the semi-arid regions in East China and India have higher population densities. The differences of population densities in the four semi-arid regions seem to show very different impacts on anthropogenic dust emission. While the inclusion of these areas in the analysis of overall interacting patterns may lead to mixed results, we have added the description and discussion on the relationships in the four regions separately (Fig. 3) in the revised manuscript. The typical economic mode has great impact on the relationship between anthropogenic dust and population densities over different semi-arid regions. The comparison in East China, India, North America, and North Africa (Fig. 3) demonstrate the Indian semi-arid region with a traditional agriculture has a close relationship between population density and anthropogenic dust. Related with other semi-arid regions, India as a developing country, agriculture is its major industry, the relationship between human activities and population is more direct, and its agriculture is an industry that directly impacts the land that is easily leading to production of anthropogenic dust. It illustrated that anthropogenic dust has a close relationship with development level of agriculture.

(3) Would the pattern be clearer after controlling AI index, or/and economic level/activity?

Response: We think the pattern will be clearer after controlling the economic level. This part of description and discussion has been added in the revised manuscript. As shown in Fig. 3, we can find that different semi-arid regions have inconsistent relationships between population density and anthropogenic dust, which illustrates the role of economic level in relationship between anthropogenic dust and population.

Part 2:
**Other comments:**

(1) In section 4.1, it would be preferable to use "mixed dust" instead of "combined dust" to avoid confusion, particularly when Figure 5 stacks (or combines) anthropogenic and natural dust burden from the "mixed" dust regions.

Response: Thanks for the suggestion. The "combined dust" has been replaced by "mixed dust," and we have checked the whole manuscript to ensure no similar problem exists in the revised manuscript.

(2) The sentence of Lines 28-29 on Page 6 can be moved to introduction section, and expressed as a key contribution of this research.

Response: Thanks for the suggestion. The sentence in lines 28-29 on Page 6 has been moved to the introduction section, as a key contribution of this research.

(3) While Figure 4 displays anthropogenic vs. combined (mixed) dust burden, the text on Page 6 talks about the natural vs. mixed dust burden. It should make them consistent.

Response: The text on Page 6 has been revised to be consistent with the figure caption.

(4) While Page 7 Line 19 says "both India and East China have higher population density (>= 250 persons km-2) which is also displayed in Figure 6, the other parts of the paper uses 45 persons km-2 for East China. Is the number in Figure 8 derived from the data of Figure 6? Please explain why.

Response: It is our poor English expression, and we have revised the text. In Line 19 on Page 7, we want to state that "For the four selected semi-arid regions, only India and East China have grids with population density greater than 250 persons km-2, most of North Africa has the population density between 10 and 40 persons km-2, and the population density in semi-arid region of North America is in the range of less than 10 persons km-2."
(5) The last paragraph of Page 7 and Figure 7 is not really relevant and could be removed. There are some contradictions in texts of the first two paragraphs on Page 8. For instance, it says 8 percentages population increase in East China in the first paragraph but 6.16 percentages in the second; 30 percentages increase in N. Africa in the first paragraph, and 29.26 percentages in the second. While the paper is generally well written, the second half of the text needs to be improved.

Response: Thanks for the suggestion. We agree with reviewer. This paragraph has been rewritten. The contradiction in text has been revised.

(6) In particular, Section 4.2 and 4.3 are not always easy to follow. For instance, what does it mean "Most semiarid regions locate in the anthropogenic dust areas" (Page 8 Line 18)? What is "rear population" (Page 8 Line 23)?

Response: Sections 4.2 and 4.3 have been revised. (1) According to the distribution of anthropogenic dust (Huang et al., 2015), anthropogenic dust not only appears in the semi-arid regions, but also relatively concentrated in the semi-arid regions. The sentence of "Most semi-arid regions are located in the anthropogenic dust areas" has been removed, in order to avoid misunderstanding. (2) "[R]ear population" should be "rare population". Similar problems no longer appear in the revised manuscript.

**Reference:**

Huang, J., Liu, J., Chen, B., and Nasiri, S. L.: Detection of anthropogenic dust using CALIPSO lidar measurements, Atmos. Chem. Phys., 15, 11653–11655, doi:10.5194/acp-15-11653-2015, 2015.
Fig. 1. Mean anthropogenic dust column burden changes as a function of population density

Fig. 2. Mean anthropogenic dust column burden changes as a function of population change

**ACPD**
Fig. 3. Anthropogenic dust probability density distributions of semi-arid regions in (a) East China, (b) India, (c) North America, (d) North Africa

**ACPD**

---

## Author Response (AR2)

We are grateful for the reviewer 2' useful advice. For the suggestion of "Just a small suggestion for the caption of Figures 13 and 14: it will be better to change "AD" to "Anthropogenic dust" because it will be easier for readers to understand what AD is in case they only want to take a glance of the figures."

**Response:** The "AD" in the caption of Figures 13 and 14 have been replaced by "Anthropogenic dust".